# Magnetic Nanoclusters Stabilized with Poly[3,4-Dihydroxybenzhydrazide] as Efficient Therapeutic Agents for Cancer Cells Destruction

**DOI:** 10.3390/nano13050933

**Published:** 2023-03-03

**Authors:** Ioana Baldea, Anca Petran, Adrian Florea, Alexandra Sevastre-Berghian, Iuliana Nenu, Gabriela Adriana Filip, Mihai Cenariu, Maria Teodora Radu, Cristian Iacovita

**Affiliations:** 1Department of Physiology, Iuliu Hatieganu University of Medicine and Pharmacy, Clinicilor 1–3 Str., 400012 Cluj-Napoca, Romania; 2National Institute for Research and Development of Isotopic and Molecular Technologies, 67–103 Donat Str., 400293 Cluj-Napoca, Romania; 3Department of Cell and Molecular Biology, Faculty of Medicine, Iuliu Hațieganu University of Medicine and Pharmacy, Pasteur 6 Str., 400349 Cluj-Napoca, Romania; 4Department of Biochemistry, University of Agricultural Sciences and Veterinary Medicine, Calea Manastur 3–5 Str., 400658 Cluj-Napoca, Romania; 5Department of Pharmaceutical Physics-Biophysics, Faculty of Pharmacy, “Iuliu Hatieganu” University of Medicine and Pharmacy, 6 Pasteur Str., 400349 Cluj-Napoca, Romania

**Keywords:** poly[3,4-dihydroxybenzhydrazide], polydopamine analogues, solvothermal method, magnetic nanoclusters, magnetic hyperthermia, apoptosis, membrane pathway, p53 pathway, oxidative stress

## Abstract

Magnetic structures exhibiting large magnetic moments are sought after in theranostic approaches that combine magnetic hyperthermia treatment (MH) and diagnostic magnetic resonance imaging in oncology, since they offer an enhanced magnetic response to an external magnetic field. We report on the synthesized production of a core–shell magnetic structure using two types of magnetite nanoclusters (MNC) based on a magnetite core and polymer shell. This was achieved through an in situ solvothermal process, using, for the first time, 3,4-dihydroxybenzhydrazide (DHBH) and poly[3,4-dihydroxybenzhydrazide] (PDHBH) as stabilizers. Transmission electron microscopy (TEM) analysis showed the formation of spherical MNC, X-ray photoelectronic spectroscopy (XPS) and Fourier transformed infrared (FT-IR) analysis proved the existence of the polymer shell. Magnetization measurement showed saturation magnetization values of 50 emu/g for PDHBH@MNC and 60 emu/g for DHBH@MNC with very low coercive field and remanence, indicating that the MNC are in a superparamagnetic state at room temperature and are thus suitable for biomedical applications. MNCs were investigated in vitro, on human normal (dermal fibroblasts-BJ) and tumor (colon adenocarcinoma-CACO2, and melanoma-A375) cell lines, in view of toxicity, antitumor effectiveness and selectivity upon magnetic hyperthermia. MNCs exhibited good biocompatibility and were internalized by all cell lines (TEM), with minimal ultrastructural changes. By means of flowcytometry apoptosis detection, fluorimetry, spectrophotometry for mitochondrial membrane potential, oxidative stress, ELISA-caspases, and Western blot–p53 pathway, we show that MH efficiently induced apoptosis mostly via the membrane pathway and to a lower extent by the mitochondrial pathway, the latter mainly observed in melanoma. Contrarily, the apoptosis rate was above the toxicity limit in fibroblasts. Due to its coating, PDHBH@MNC showed selective antitumor efficacy and can be further used in theranostics since the PDHBH polymer provides multiple reaction sites for the attachment of therapeutic molecules.

## 1. Introduction

In biomedicine and life sciences, development of nanomaterials based on magnetic nanoparticles (MNPs) is currently one of the most active research fields with some of the fastest growth [1,2,3,4]. In particular, nanomaterials with a magnetic core–shell architecture, are often preferred due to the peculiar magnetic properties provided by the core along with the enhanced biocompatibility and specific functionality ensured by the protective shell [5,6,7]. Due to the bi-functionality of these types of nanostructures, they have been successfully used for the highly efficient labeling of human stem cells [8,9], as magnetic carriers for drug delivery [10,11,12], as fluorescent intracellular probes [13], as therapeutic agents in photodynamic therapy [14,15,16], and as magnetic hyperthermia (MH) applications [17,18,19,20,21,22].

The high-performance biomedical applications demand a magnetic core exhibiting a large magnetic moment and no residual magnetization in the absence of an external magnetic field to avoid the aggregation of MNPs and blood vessel embolization [23,24]. This property is called superparamagnetism (SP) and restricts the dimension of the magnetic core to several tens of nanometers [25,26]. Thus, individual SP-MNPs display limited values of their magnetic moments that hamper their successful implementation in magnetic field assisted biomedical applications. However, when individual SP-MNPs are self-assembled or self-aggregated in clusters of several hundreds of nanometers, the magnetic interaction between them can activate higher magnetic moments, enhancing the magnetic responses of the formed clusters [27,28,29,30,31]. Moreover, it has been shown that the collective behavior of nanoparticles’ magnetic moments within clusters under an alternating external magnetic field can lead to an important increase of the heating capabilities in MH experiments [28,32,33].

The formation of magnetic nanoclusters (MNC) has been achieved by the assembly of pre-synthesized MNPs through colloidal (solvo-phobic) interactions mediated by ligands [34,35]. It has been shown that a direct aggregation of MNPs in situ during synthesis is also possible resulting in an MNC with a well-defined shape and controlled morphology [36,37,38,39]. Among different synthesis methods employed in the elaboration of MNC, the solvothermal/hydrothermal method is one of the most reproducible in terms of size or shape, by adjusting the reagent parameters [40,41,42,43,44,45]. In addition, the solvothermal/hydrothermal method allows functional coating of MNPs directly in situ by adding reactants in the reagent mixture, thus giving rise to core–shell nanostructures in one single step. Since the in vitro and in vivo studies, and later clinical trials, require large quantities of MNC, this capability represents an important advantage over the post-synthesis coating of MNC, which implies the accomplishment of further steps until the final product is reached.

Dopamine and L-DOPA are examples of molecules that act as surfactants, forming a shell around MNC, either by means of in situ synthesis [46] or through a post-functionalization process [47,48]. The class of catecholamines has received special attention in recent years, especially due to the easy process of obtaining the corresponding polymers but also because of the variety its of applications [49]. Not all derivatives in this class polymerize so easily, e.g., 3,4-dihydroxybenzhydrazide (DHBH). Though this monomer contains a catecholic ring and a terminal amine, which are responsible for functions in the surface adhesion process, the hydrazine group involved in the polymerization process significantly influences the surface adhesion capacity [50]. Despite the fact that this monomer is not as versatile as its counterparts, the presence of multiple functions (the catecholic ring or the hydrazine group) make it of great interest as a surfactant in the solvothermal process that is involved in obtaining the respective MNC and that is probably also involved in the presence of Fe^2+^/Fe^3+^ ions in the thermic process, and thus helps form the corresponding poly-dihydroxybenzhydrazide (PDHBH) as a coating for the MNC. In comparison with other polycatecholamines, where the monomer is soluble and the corresponding polymer is not, our polycatecholamine involves a totally opposite process, thus requiring us to enhance the solubility of our PDHBH as a special feature. Another reason that this PDHBH is a good candidate in different coatings is its complex structure based on different oligomeric units with functionalities, such as catecholic ring, hydrazine, acylhydrazine, acylhydrazone or 1,3,4-oxadiazoline, which provide multiple reaction sites attached to the core and others at the top of the polymeric shell. Based on these aspects we provide PDHBH as a new coating for MNC in a solvothermal process.

Magnetic hyperthermia, a localized form of hyperthermia, is based on the property of MNPs to convert the electromagnetic energy generated by an alternating magnetic field (AMF) into heat [51]. Firstly, magnetic nanoparticles are internalized by the cells in intracellular vesicles, such as endosomes or lysosomes [52]. Then, the tumor is exposed to AMF and the resulting heat leads to apoptotic or necrotic cell death within the tumor site. Several clinical trials have demonstrated its usefulness without causing severe side effects to healthy surrounding tissues [53]. Moreover, after exposure to AMF, hot spots are produced inside the cell, leading to non-homogeneous heating, without alterations of the global temperature. Magnetic nanoparticles induce lysosome permeabilization with consequent release of the lysosome proteolytic content to the cytosol and eventually cell death [54]. Tumor cell death also activates immune pathways that might lead to far-reaching antitumor results. Favorable results have been shown by different magnetic mediators, such as ferromagnetic alloy thermoseeds and metallic stents for interstitial implant-based hyperthermia and iron oxide MNPs [55]. Iron oxide MNPs are most commonly used for MH, given their physical and chemical properties [56,57]. Moreover, it seems that MH mediated by iron oxide MNPs, associated with conventional treatments, did not induce severe adverse effects, making them a possible niche to explore in oncology [56,57].

Herein, we report the first approach regarding the ability of DHBH and PDHBH to act as new stabilizers for MNC (DHBH@MNC and PDHBH@MNC) after an in situ solvothermal process. The coated MNC were fully characterized using transmission electron microscopy (TEM), X-ray photoelectron spectroscopy (XPS), infrared spectroscopy (FT-IR) and vibrating sample magnetometry (VSM) while their heating capabilities were investigated by a magnetic hyperthermia system. A first step for applications in biomedicine is to demonstrate their biocompatibility in vitro, therefore in the current study the MNC were tested against human cell lines of normal dermal fibroblasts (BJ), tumor colon adenocarcinoma (CACO2) and melanoma (A375). The potential of the MNC to induce selective tumor cell death in vitro by means of MH was analyzed comparatively on the cancerous and normal cell lines, by means of viability assays (MTT, LDH), MNC uptake (TEM), cell death–apoptosis/necrosis ratio (flowcytometry) mitochondrial membrane potential, oxidative stress (fluorometry, spectrophotometry) caspases level (ELISA), and p53 pathway (Western blot).

## 2. Materials and Methods

### 2.1. Materials

Iron (III) chloride hexahydrate (FeCl_3_·6H_2_O), sodium acetate anhydrous (CH_3_-COONa, NaAc) and 3,4-dihydroxybenzhydrazide (DHBH) were purchased from Sigma-Aldrich (Heidelberg, Germany), while PDHBH was obtained after the procedure described in [50]. Ethylene glycol (EG) and diethylene glycol (DEG) were provided by Alfa Aesar. All chemicals were of analytical grade and used without further purification.

### 2.2. Synthesis Method

The MNC coated with either DHBH or PDHBH were obtained by solvothermal method according to the following protocol: FeCl_3_·6H_2_O (7.3 mmol, 2 g) and NaAc (36 mmol, 3 g) were dissolved in a 40 mL mixed solvent of DEG and EG (50%:50%, *v*:*v*) to form a clear solution under mechanical stirring at 60 °C for 3 h. Separately, DHBH (1.78 mmol, 0.3 g) and PDHBH (0.15 g) were suspended in 20 mL mixture of DEG and EG (50%:50%, *v*:*v*) at room temperature for 2 h. After that, the three solutions were mixed together in a Teflon-lined stainless-steel autoclave (100 mL volume) and then sealed and heated at 200 °C. After 14 h reaction time, the autoclave was cooled to room temperature. The dark suspension was washed several times with a methanol–water mixture and distilled water. The magnetic clusters were kept in 10 mL of water, as a suspension, and part of the sample was dried and used for different analyses.

### 2.3. Characterisation Methods

A JEOL JEM-100CX II (JEOL, Tokyo, Japan) transmission electron microscope working at 80 kV and equipped with a MegaView G3 camera (Emsis, Münster, Germany) running with Radius 2.1 software (Emsis) was used to visualize the MNC. A drop of water suspension of MNC (10 µg_MNPs_/mL) was deposited on a carbon-coated copper grid, and after 5 min the excess water was removed with a filter paper.

The surface chemical composition of the samples was investigated by X-ray photoelectron spectroscopy (XPS) using a spectrometer SPECS equipped with an Al/Mg dual-anode X-ray source, a PHOIBOS 150 2D CCD hemispherical energy analyzer, and a multichanneltron detector with vacuum maintained at 1 × 10^−9^ Torr. The Al K_α_ X-ray source (1486.6 eV) was operated at 200 W. The XPS survey spectra were recorded at 30 eV pass energy and 0.5 eV/step. The high resolution spectra for the individual elements (Fe, C, O, N) were recorded by accumulating 10 scans at 30 eV pass energy and 0.1 eV/step. Data analysis and deconvolutions was performed using CasaXPS software with a Gaussian–Lorentzian product function and a nonlinear Shirley background subtraction. Peak shifts due to any apparent charging were normalized with the C 1 s peak set to 284.8 eV.

FTIR spectra were carried out on a JASCO FTIR 610 spectrophotometer in transmission configuration in the range 4000–400 cm^−1^ and spectral resolution 4 cm^−1^ using KBr pellet technique.

Magnetic measurements were performed on powder samples at room temperature using a vibrating sample magnetometer (VSM) from Cryogenics (London, UK).

### 2.4. Cell Cultures

The assays were performed on three human cell lines: one normal line of dermal human fibroblasts (BJ, ATCC CRL-2522^™^) and two cancer cell lines, a Caucasian colon adenocarcinoma CACO2 (ECCAC, Sigma Aldrich, Co., Heidelberg, Germany), and a metastatic melanoma A375 (ATCC^®^ CRL-1619™) from ATCC (Gaithersburg, MD, USA). Cells were cultured in Dulbecco’s modified Eagle medium (DMEM) supplemented with 5% fetal calf serum, 50 µg/mL gentamicin and 5 ng/mL amphotericin, all from Biochrom AG (Berlin, Germany). Cultures were fed twice weekly.

### 2.5. Viability Assay

Cell survival was assessed through the colorimetric measurement of formazan, a colored compound synthesized by viable cells, using CellTiter 96^®^ AQueous Non-Radioactive Cell Proliferation Assay (Promega Corporation, Madison, WI, USA). The cells-FB, CACO2 and A375 were cultivated at a density of 10^4^/wells in 96-well plaques (TPP, Transadingen, Switzerland) for 24 h, then exposed for 24 h to DHBH@MNC and PDHBH@MNC in concentrations ranging between 0–200 µg/mL, suspended in medium. Viability was measured calorimetrically, using an ELISA plate reader (Tecan, Männedorf, Switzerland) at 540 nm. All the experiments were completed in triplicate. Untreated cell cultures were used as controls. Results are presented as % of untreated control, the dose that caused a viability decrease below 70% was considered toxic. Based on these results, the dose selected for hyperthermia experiments was 50 µg/mL.

### 2.6. Transmission Electron Microscopy

The FB, CACO2 and A375 cells were incubated for 24 h with DHBH@MNC and PDHBH@MNC in concentrations of 50 µg/mL, and were, with their corresponding controls, exposed to normal medium and processed for transmission electron microscopy (TEM) examination. Briefly, after tripsinization, the cells were resuspended into an ice-cold 2.7% glutaraldehyde (Electron Microscopy Sciences, Hatfield, PA, USA) solution in 0.1 M phosphate buffer (pH 7.4) and centrifuged for 5 min at 500× *g*. The fixative was changed and prefixation continued for 1.5 h at 4 °C. The cells were next washed four times (1 h each at 4 °C) with 0.1 M phosphate buffer (pH 7.4), and post-fixation was performed for 1.5 h at 4 °C with 1.5% osmium tetroxide (Sigma-Aldrich, St. Louis, MO, USA) solution in 0.15 M phosphate buffer (pH 7.4). The samples were next dehydrated in ethanol series (30–100%, 30 min each), and infiltrated with ethanol solutions of EMBed-812 (Electron Microscopy Sciences) of increasing concentrations (30%, 50%, 70% 1 h each and 3 × pure resin, 12 h each) at room temperature. Sections of 60–70 nm were cut with a DIATOME diamond knife (Hatfield, PA, USA) on a Bromma 8800 ULTRATOME III (LKB, Stockholm, Sweden) and were collected on 300 mesh copper grids (Agar Scientific Ltd., Stansted, UK) and contrasted for 5 min with 13% ethanol solution of uranyl acetate (Merck, Billerica, MA, USA). Sections were examined with a JEM 100CXII transmission electron microscope (Jeol, Tokyo, Japan) at 80 kV equipped with a MegaView G3 camera controlled by a Radius 2.1 software (both from Emsis, Münster, Germany).

### 2.7. In Vitro Magnetic Hyperthermia

Cells cultivated in Petri dishes (Ø = 9 cm) at a density of 4 × 10^4^/cm^2^ were exposed to DHBH@MNC and PDHBH@MNC in concentrations of 50 µg/mL for 24 h, then washed and collected by trypsinization. The cell pellet (approx. 2.5 × 10^6^ cells) was resuspended in 400 µL of fresh medium. The cellular suspension was equally divided into two aliquots of 200 µL each. One of the aliquots was kept in a water bath at 37 °C (negative control), while the other aliquot was exposed to an alternating magnetic field (AMF) for 40 min, working at a fixed frequency of 355 kHz and amplitude of 40 kA/m. The cells were placed in an Eppendorf tube in the middle of an 8-turn coil connected to a commercially available magnetic hyperthermia system, the Easy Heat 0224 from Ambrell (Scottsville, NY, USA). The tube was surrounded by plastic pipes connected to a Peltier element and thermostatted at 37 °C. The temperature was measured with an optical fiber temperature sensor (0.1 °C accuracy) placed in the middle of the cellular suspension volume. Following magnetic hyperthermia (MH) exposure, cells were immediately resuspended in fresh medium and cultivated either on 96-well plaques or on Petri dishes for an additional 24 h, then collected and processed according to the specific protocol. Cell viability following MH exposure was measured as described above.

### 2.8. ELISA

Caspase 3, 8 and 9 ELISA Immunoassay kits from R&D Systems, Inc. (Minneapolis, MN, USA) were used according to the manufacturer’s instructions; readings were taken at 450 nm with correction wavelength at 540 nm, using an ELISA plate reader (Tecan).

### 2.9. Cells Lysis

Cell lysates were prepared as described [58]. Protein concentrations were determined by the Bradford method (Biorad, Hercules, CA, USA) and using bovine serum albumin as standard. For all assays, the lysates were corrected by total protein concentration.

### 2.10. Spectrophotometry and Fluorometry

Quantification of malondialdehyde (MDA), a marker for the peroxidation of membrane lipids induced by oxidative damage, was undertaken via spectrophotometry, from cell lysates. Data were expressed as nanomoles/mg protein [58,59]. All reactives were purchased from Sigma. Amplex™ Red Hydrogen Peroxide/Peroxidase Assay Kit (Invitrogen, Fisher Thermoscientific, Waltham, MA, USA) was used to measure production of H_2_O_2_, according to the manufacturer’s instructions. Readings were taken via fluorometry—excitation 571 nm, emission 590 nm, intervals 0 h, 0.5 h, 1 h, 1.5 h, 2 h—immediately following MH exposure, using a SpectraMax iD3 Multi-Mode Microplate Detection Platform (Molecular Devices, San Jose, CA, USA). Data are expressed as OD590.

### 2.11. Western Blotting

Lysates (20 µg protein/lane) were separated by electrophoresis on SDS PAGE gels and transferred to polyvinylidene difluoride membranes, using Biorad Miniprotean system (BioRad). Blots were blocked and then incubated with antibodies against TP53 tumor suppressor protein (p53), B cell lymphoma 2 (BCL-2), and BCL-2 associated X protein (BAX) (Santa Cruz Biotechnology, Santa Cruz, CA, USA), then further washed and incubated with corresponding secondary peroxidase-linked antibodies (Santa Cruz Biotechnology). Proteins were detected using Supersignal West FemtoChemiluminiscent substrate (Thermo Fisher Scientific, Rockford, IL, USA), and a Gel Doc Imaging system equipped with an XRS camera and Quantity One analysis software Image Lab 6.0 Biorad Laboratories). βactin (Sigma Aldrich) was used as a protein loading control.

### 2.12. Statistical Analysis

The statistical differences between experimental materials and control groups were evaluated by two-way ANOVA followed by Bonferroni post-test, Student’s t Test, and Kruskal–Wallis test for Western blot assessment. All the values in text and figures are expressed as mean ± standard deviation, all experiments were conducted in triplicate; results were considered significant for *p* ≤ 0.05. Statistical package used for data analysis was Prism version 4.00 for Windows (GraphPad Software, San Diego, CA, USA).

## 3. Results and Discussion

### 3.1. Magnetic Nanoclusters Characterization

The morphology of the as-synthesized MNC were characterized by TEM in order to confirm the formation of spherical magnetic clusters with a narrow size distribution, as shown in Figure 1. The MNC produced with the stabilizing agent DHBH exhibited a well-defined spherical shape. The diameter of DHBH@MNC, determined by measuring more than 500 MNC based on TEM images and fitted using the log-normal distribution function, showed a broad size distribution with an average value of 175 nm. The high resolution TEM image revealed that the individual spherical MNC were composed of small multicore MNPs, which proves the efficient nanoclustering of MNPs by using both coating agents. Spherical MNC with multicore architecture are also produced by using PDHBH as stabilizing agent. This time, the size distribution was narrow, exhibiting a smaller mean diameter of 135 nm, which might be a consequence of the better solubility of the polymer vs. monomer in a homogeneous reaction environment.

XPS analysis was used to demonstrate the successful preparation of magnetic nanoclusters coated with poly[3,4-dihydroxybenzhydrazide]. The high resolution XPS spectra of N1s were deconvoluted into components in order to gain an insight into the surface chemistry of the prepared samples (Figure 2). The N1s high resolution spectra of initially prepared sample show a peak at 399–399.4 eV attributed to N-C, NH_2_ and components at higher binding energies which correspond to N-(C=O)- (399.6–400eV) and NH_3_ (401.7–401.8 eV). The types of N found in both materials confirm the polymer coating, with the small exception that DHBH@MNC provides the more primary and only quaternary amines that might result from a different arrangement of DHBH in the solvothermal process. Appendix A presents the binding energies determined from the N1s signals and the ratio of the identified chemical states of the surface components for both DHBH@MNC and PDHBH@MNC.

The formation PDHBH coating for both DHBH@MNC and PDHBH@MNC was confirmed by FTIR spectroscopy as expected due to the capacity of Fe^3+^ ions to catalyze polycatecholamines [60] e.g., DHBH. Figure 3 shows the overlapped spectra of simple PDHBH polymer and MNC obtained in the solvothermal process. The PDHBH present in both materials is confirmed by the similarities of the bulk polymer and the coated MNC with specific bonds at 1030–1300 cm^−1^ for C-N, C-C, and C-O; 1450–1700 cm^−1^ for C=C, C=N, and C=O; and at 2800–3600 cm^−1^ for C-H, N-H, and O-H. [50] The vibrational bands of magnetite correspond to 582 and 634 cm^−1^.More specifically, 582 cm^−1^ corresponds to the vibration of Fe^2+^-O^2^ bond and 634 cm^−1^ is due to the symmetry degeneration on octahedral B sites [61,62].

The magnetic properties of both types of MNC were measured using a vibrating sample magnetometer (VSM) at room temperature, Figure 4. From the magnetization versus applied field curve, the saturation magnetization (Ms) was measured. The DHBH@MNC displayed a saturation magnetization (M_s_) of 60 emu/g which is in accordance with other studies on MNC synthesized by the solvothermal method [29,40,43,44]. The PDHBH@MNC exhibited a lower M_s_ of 50 emu/g, due most probably to their smaller diameter as observed in TEM and the greater thickness of the PDHBH shell when compared with DHBH. Generally, small nanoparticles have a large surface area and thus the effect of spin disorder at the surface is high, resulting in small Ms and high magnetic anisotropy. This effect has been reported in a large number of previous studies [63,64,65]. In addition, considering the core–shell structure, the surface layer of nanoclusters without magnetic ordering may also have an important contribution to the obtained results.

For both types of MNC, hysteresis behavior is not present, and the coercive field (H_c_) and remanence (M_r_) are very low, indicating that the MNC are in a superparamagnetic state at room temperature [31], thus making them suitable for biomedical applications.

### 3.2. Cell Viability Assay

Several studies have reported a decreased susceptibility to MH of normal cells compared with tumor cells [57,66,67], due to various differences such as the aerobic versus anaerobic cellular metabolism, decreased mitotic index, increased oxygen supply, a normal versus a tumor microenvironment, and adaptation to hyperthermia, all of which render the tumor cells more susceptible to apoptosis and/or enhance their response to radio/chemotherapy.

However, in vitro MH may be able to induce a toxic effect on normal cells, due to the combination of the cytotoxicity of the MNPs and heat release by MNPs. Therefore, in our study we have comparatively used a human cell line of normal dermal fibroblasts (BJ) and two tumor cell lines (colon adenocarcinoma (CACO2) and Caucasian metastatic melanoma (A375)) to firstly assess the cytotoxicity of MNC and subsequently their in vitro MH capabilities to induce cell death.

Both types of MNC showed no toxic effects up to the concentration of 100 µg/mL in all cell lines (Figure 5). By doubling the concentration of MNC to 200 µg/mL, the cell viability suffers considerable changes. For instance, a decrease in cell viability below the toxicity limit of 70% of untreated controls was induced by the PDHBH@MNC in all cell lines. On the contrary, the DHBH@MNC are less toxic: the cell viability decreased close to 70% for BJ and CACO2 cell lines, while on A375 cell lines it was around 80%.

### 3.3. Cellular Uptake and Cell Alterations

The intracellular presence of both types of MNC in the cells, as well as their cellular impact effects were evaluated by TEM examination.

Control fibroblasts (BJ) displayed large euchromatic nuclei with prominent nucleoli, abundant rough endoplasmic reticulum, and many glycogen granules. Rare vacuoles were observed in the cytoplasm, and many filopodia were visible at the periphery (Figure 6A,B).

In some cells 1–2 mitochondria were visible (not shown). Incubation with both types of MNC resulted in their internalization to considerable amounts (Figure 6C–F). Thus, MNC were found both inside the endosomes and dispersed freely into the cytoplasm either as single particles (not shown), or as aggregates (Figure 6C,D). The cellular ultrastructure remained generally unchanged after incubation with these nanoparticles, the only identified reaction being the presence of a higher number of cytoplasmic vacuoles (possibly endosomes). PDHBH@MNC showed a similar behavior inside the cells, having the same cellular distribution as DHBH@MNC. The cellular organelles were not visibly affected (Figure 6E,F), but many autophagosomes were identified (Figure 6F) and a number of vesicles were found in some of the examined cells (not shown).

Control CACO2 colon cancer cells presented filopodia at the level of their plasma membrane. They contained predominantly euchromatic nuclei, with prominent nucleoli, and in cytoplasm many large oval mitochondria were found, along with important amounts of glycogen and few vacuoles (Figure 7A,B). After exposure, both types of MNC were found in high numbers in their abundant cytoplasm, even in proximity to the nucleus (Figure 7C,D). No MNC were found in endosomes. Apart from cytoplasmic vacuolation in rare cells (not shown), only the presence of autophagosomes and a visible decrease in the number of glycogen granules were noted (Figure 7C). Many dividing cells containing clusters of DHBH@MNC in their cytoplasm were found (not shown). PDHBH@MNC were internalized in high numbers, but they were observed both inside endosomes (Figure 7E) and as clusters or dispersed into the cytosol (Figure 7F). Additionally, vacuoles were present in some cells (Figure 7E) and a moderate reduction of glycogen (Figure 7E,F) was obvious in most of the examined cells in this group.

Control A375 melanoma cells appeared as smaller cells as compared with fibroblasts, with an approximately round shape and with many thin filopodia (Figure 8).

They contained very large, indented and euchromatic nuclei, with large nucleoli. In the cytoplasm, only a few mitochondria were identified as well as several profiles of rough endoplasmic reticulum and rare vacuoles; glycogen granules were present in high number (Figure 8A,B). In the melanoma cells, the DHBH@MNC and PDHBH@MNC were found to be more or less dispersed into the cytoplasm in all cases (Figure 8C–E). No MNC were found located in endosomes. No cellular effects of the internalized nanoparticles were identified, excepting for cytoplasmic vacuolation in a small number of the examined cells (not shown). In all types of cells there was a tendency of MNC to disintegrate either within the endosomes or after they arrived in contact with the fluid phase of the cytoplasm (insets in (Figure 6C,E, Figure 7D,E and Figure 8D,F)).

Several studies have reported that the iron oxide MNPs and MNC can be degraded and fragmented, in the lysosomes, following their cellular uptake in dendritic cells [68], GL261 mouse glioma cells [69] and different types of cancer cells [43]. Exposure of cells to iron oxide MNPs can cause alterations in cell morphology, or even loss of cell integrity. In HMEC and HUVEC cells, increased intracellular gap junctions were found, signaling an impaired endothelial barrier, dysfunction of the plasma membrane and cytoskeleton impairment. In CACO2, transepithelial electrical resistance (TEER) measurement showed loss of intracellular junctional complexes [70]. When used in vivo, in high doses, iron oxide MNPs coated with gold (30 mg/kg), were reported to induce changes such as vacuolization and intracytoplasmatic swollen fat globules in the liver, or vacuoles in mithochondria, cytoplasm and lysosomes of kidney, heart and lung tissue [71].

### 3.4. In Vitro Magnetic Hyperthermia

Based on the viability results (Figure 5), and our previous experience on in vitro MH [66], a concentration of 50 µg/mL has been chosen for the in vitro MH experiments. Thus, the cell cultures from the three lines, previously incubated with both types of MNC for 24 h, were exposed for 40 min to an AMF of fixed amplitude (40 kA/m) and frequency (355 kHz). The MH-induced cytotoxicity was evaluated with respect to a control sample by performing viability assays, LDH membrane integrity assay and mitochondrial potential measurement at 24 h after MH exposure. In the first step, untreated normal and cancer cells were exposed to AMF. A modest increase in sample temperature has been recorded, not exceeding 1 °C (Appendix A), which did not affect cellular viability. On the contrary the heating curves of all cells (normal and cancer) containing internalized MNC, exhibited a relevant increase in the temperature (3.5–3.6 °C) in the first 15 min followed by the formation of a plateau (Appendix A). The saturation temperature (around 40.5–40.6 °C) is almost identical for both types of MNC and all three types of cells. Fibroblasts showed a slight viability decrease, especially in the case of DHBH@MNC, but viability was still above the toxicity limit of 70% (Figure 9). LDH level was increased compared with the control, but not significantly for both types of MNC. DHBH@MNC showed a stronger effect compared with PDHBH@MNC (Figure 9). MMP was slightly decreased upon MH treatment, but again not significantly. In other words, the BJ cells incubated with MNC are resistant to MH treatment. The situation is completely different for CACO2 cells, which showed a strong decrease in viability upon MH treatment with both types of MNC. PDHBH@MNC exhibited a stronger effect, decreasing the cell viability up to 41.25%. The LDH was significantly increased for both types of MNC, with the maximum level recorded for PDHBH@MNC. MMP detected a toxicity as well, which is more pronounced for cells incubated with PDHBH@MNC. The cytotoxicity of melanoma A375 cells decreased below the 70% toxicity limit only for PDHBH@MNC. Instead, the LDH level was strongly increased for both types of MNC. Similar to CACO2 cells, the PDHBH@MNC showed a pronounced effect. MMP was significantly decreased by MH treatment; both types of MNC inducing similar effects. Therefore, the normal BJ cells incubated with MNC are more resilient to MH treatment than cancer cell lines, which were considerably affected, as also demonstrated in other studies [66].

### 3.5. Cell Death Mechanism

Flowcytometric assessment of the cell death using annexin V/PI staining shows a very good survival in all cells exposed to either MNC, or MNC and AMF, which is consistent with the cytotoxicity assay, LDH and MMP results (Appendix A). In all cells, the mechanism of MH-induced cell death was early apoptosis, but the level of apoptosis was very different depending on the cell type: fibroblast10.2% for DHBH@MNC and 7.5% for PDHBH@MNC; CACO2 28.1% for DHBH@MNC and 38.3% for PDHBH@MNC; and A375 32.5% for DHBH@MNC and 56.1% for PDHBH@MNC. Overall, data are consistent with the viability and the LDH assay. Since early apoptosis (annexin+/PI- cells) was seen as late as 24 h following MH exposure, it could be possible that the maximum extent of the MH effects needs more time to develop, therefore, in future studies, a longer follow-up of the treated cells might be useful to assess the efficacy of the MH. Indeed, in a study on CACO2 and MCF-7 cells treated with iron oxide MNPs functionalized with carboxymethyl dextrane (IO-CMDX) in a concentration of 0.6 mg_MNPs_/mL, and exposed to an AMF (20 kA/m, 238 kHz) for 1 h and 2 h, reaching a temperature of 42–45 °C, showed a peak cytotoxicity at 48 h after MH treatment within 2 h exposure [72,73].

The levels of caspases were measured by ELISA (Figure 10) to assess the type of apoptotic cell death induced by the MH. Caspase 3, involved in the common apoptosis pathway was significantly increased by MH treatment in all groups. In fibroblasts, caspase 3 level was at minimum, compared with the other cells and DHBH@MNC induced the strongest effect. In cancer cells, caspase 3 showed a more important increase, the maximum level was reached for PDHBH@MNC-mediated MH. In fibroblasts, caspase 8, involved in extrinsic apoptosis was only significantly increased by DHBH@MNC-mediated MH. In CACO2 and A375 cells, the level of caspase 8 was significantly increased in all groups incubated with MNC, while MH treatment had a significant impact compared with MNC alone. PDHBH@MNC induced the highest caspase 8 increase. Caspase 9, involved in mitochondrial apoptosis was increased in fibroblasts by DHBH@MNC w/wo MH treatment, without significance between the two groups. This suggests that caspase 9 induction is mostly dependent on the presence of the MNC, rather than the MH exposure. In CACO2, there was an increase of caspase 9 in PDHBH@MNC groups w/wo MH treatment; MH exposure slightly increased caspase 9 compared with PDHBH@MNC alone, without a significant difference between the two groups. In A375, caspase 9 was slightly decreased by MNC incubation and MH exposure showed no additional effect. These results suggest that the main mechanism of apoptosis induction following DHBH@MNC/PDHBH@MNC-mediated MH treatment is the extrinsic apoptosis, mediated by the activation of caspase 8 and, to a lesser extent, mitochondrial apoptosis. Data are also sustained by the limited MMP decreases, compared with the extent of apoptosis, shown by the flowcytometry evaluation.

In the literature, both types of apoptotic death are described as being induced by MH treatment. In a recent work, Beola et al. [74] showed that the mechanism of cell death induced by MH exposure depends on the concentration of MNPs using a 3D cell culture model of a highly phagocytic macrophage cell line. A low concentration of SP–iron oxide MNPs (SPIONs) led to an increased BAX/BCL2 ratio and favored apoptosis by the mitochondrial pathway, while the exposure to a high SPIONs concentration led to the increase of caspase 8 without BAX/BCL2 increase, probably leading to extrinsic apoptosis. Both mechanisms can be activated at the same time by MH-triggered signals such as p53 activation, DNA damage, hypoxia, survival factor deprivation for the intrinsic pathway activation, cell microenvironment induced cytotoxic stress, and/or the expression of the death receptors CD95 (APO-1/Fas), TNF receptor 1 (TNFRI), TNF-related apoptosis-inducing ligand-receptor 1 (TRAIL-R1), and TRAIL-R2 on cell surface activated by their specific ligands (CD95 ligand (CD95L), TNFα, lymphotoxin-α, and TRAIL) [75]. Iron oxide MNPs functionalized with 3-chloropropyltrimethoxysilane (CPTMOS) and conjugated by 1-((3-(4-chlorophenyl)-1-phenyl-1H-pyrazole-4-yl)methylene)-2-(4-phenylthiazol-2-yl) hydrazine (TP) (Fe_3_O_4_@CPTMOS/TP NPs) activated the intrinsic pathway in gastric cancer cells [76]. Others reported that MNPs activate both apoptotic pathways as seen in pancreatic BxPC-3 cells using gemcitabine-conjugated MNPs and hyperthermia [77].

Many mitochondria-targeted nanoparticles have been synthesized for chemotherapy, photothermal therapy (PTT), photodynamic therapy (PDT), chemodynamic therapy (CDT), sonodynamic therapy (SDT), radiotherapy (RDT) and immunotherapy to enhance their anti-tumor efficacy (reviewed in Gao [78]). The aim was to trigger mitochondrial apoptosis and/or necrosis, by inducing damage to the mitochondrial DNA or oxidative stress. In PTT, the increase of the temperature in the mitochondrial area associated with the decrease of the heat shock proteins secretion can lead to mitochondrial cell death, while in PDT, SDT or RDT, the ROS generated by exposure to light, ultrasounds or ionizing radiation can lead to the same effect. There are many drawbacks of these procedures that limit their efficacy, such as treatment toxicity, low penetration in the solid tumors for PDT, or the increased secretion of antiapoptotic factors that provide the cancer cells the means to escape therapy.

In the apoptosis induction by the mitochondrial pathway, the mitochondrial permeability transition pore (PTP) protein complex plays a central role. PTP is located between the outer and the inner mitochondrial membrane and is primarily a voltage dependent anion channel at the outer membrane and an adenine nucleotide translocase at the inner membrane. PTP activation leads to increased permeability of the mitochondrial membranes, leading to the loss of the mitochondrial potential and subsequent large opening of the PTP to allow the cytochome to be released into the cytoplasm and initiate caspase activation. The process can be activated by BAX pro-apoptotic protein released by mild heat exposure (40 °C) in bovine mammary epithelium [79]. Hyperthermia (43 °C exposure) has also been shown to activate mitochondrial apoptosis in mice embryos [80] and human osteosarcoma cells [81], however, in the latter, the cells also showed endoplasmic reticulum (ER) stress.

The induction of several mechanisms depends on the internalization of the SPIONS, but also on their physicochemical properties and especially their heating capability. In our experimental setting, temperature increases during MH were moderate and similar in all cells tested. However, the MH exposure still induced selective cytotoxic effects in the tumor cells, especially in the case of cells incubated with PDHBH@MNC. Other studies have also demonstrated that efficient MH does not necessarily need a high temperature increase of the sample volume [82]. A high, localized intracellular increase in temperature has been reported near the MNPs upon AMF stimulation, but the temperature increase was dissipated with distance due to the high viscosity of the intracellular medium, which does not allow heat transfer [83]. The MH treatment (499 kHz, 20 mT) of HeLa cells incubated with polyacrylic acid-coated IONPs at a concentration of 50 µg/mL (diameter < 20 nm), strongly increased the local temperatures inside the cell, especially where the concentration of the MNPs was very high, up to 70 °C [83]. Therefore, the localization of the MNPs within the cell is very important, since the MH-triggered increase of temperature seems to be a highly localized phenomenon. In our experimental setting, the MNC were found in endosomes (fibroblasts and colon cells) and also freely in the cytosol in all three cell lines. Autophagosomes were seen in fibroblasts and colon cells, but not in melanoma.

### 3.6. Oxidative Stress Assessment

There are two possible mechanisms that explain the cell death induced by SPIONS-mediated MH: lysosomal membrane permeabilization (LMP) and free radical generation [75]. The reactive oxygen species (ROS) in MH treatment can be increased because of the enzymatic degradation of the SPIONs in the lysosomes leading to the release of iron into the cytosol. Iron can then catalyze, through the Fenton reaction, the formation of hydroxyl radicals from hydrogen peroxide. Mitochondrial disfunction may be another source of ROS because of the damage to the mitochondrial membrane, the interaction between the SPIONS and NADPH oxidase in the plasma membrane during internalization of the SPIONS or even generation of the ROS at the surface of the SPIONs, depending on their coating [75].

The level of hydrogen peroxide released by the cells was evaluated by the Amplex Red assay, starting immediately after MH exposure for a period of 2 h (Figure 11). This assay is based on the property of H_2_O_2_ to pass through the membranes into the medium, where it reacts with the Amplex Red to produce resorufin in the presence of the horseradish peroxidase [84]. H_2_O_2_ is normally produced by mitochondria and its production can be enhanced by MH, as previously reported [85,86]. MH increases transition metal ions leading to the formation of superoxide anions, which in turn are transformed into H_2_O_2_ either spontaneously or by the superoxide dismutase enzyme [86]. H_2_O_2_ is then further transformed by Fenton reaction into a short lived, but extremely reactive, hydroxyl radical [86]. Moreover, the level of MnZn SOD has been reported to be decreased by MH, probably through the heat inactivation of the enzyme. Treating cells with the catalase involved in H_2_O_2_ degradation, but not superoxide dismutase, has been reported to protect the cells against heat-induced apoptosis, which suggests that H_2_O_2_ can play an important role in the apoptosis induced by MH [87]. H_2_O_2_ can induce reversible oxidation on the Cys residues of the proteins, such as phosphatases, transcription factors, structural protein, protein kinases, and ion channels, which impairs their function [88].

In all three cell lines, there was a time-dependent increase of the H_2_O_2_ level, in all groups. In fibroblasts, H_2_O_2_ was increased by MH mediated by DHBH@MNC, starting at 1 h after exposure, while PDHBH@MNC had very little impact, being significant at 30 min. The level of malondialdehyde (MDA), a marker for lipid peroxidation generated by the presence of ROS, was only increased by DHBH@MNC-mediated MH, consistent with the Amplex Red result. In the CACO2 cells, all groups showed a similar time-dependent increase of H_2_O_2_, with no relationship to the treatment applied. Moreover, the MDA level was only slightly increased (not significantly) by MNC incubation and MH treatment. In the A375 cells, DHBH@MNC exposure w/wo MH treatment led to an increase in the H_2_O_2_ level, reaching maximum at 1 hour. There were no significant differences between MH exposure and DHBH@MNC incubation alone, which suggests that the H_2_O_2_ increase was mostly generated by the presence of the MNC and not by the MH treatment. The PDHBH@MNC-mediated MH treatment led to a strong significant increase of H_2_O_2_, when compared with the controls and the PDHBH@MNC incubated group. MDA was slightly increased by both PDHBH@MNC incubation and MH exposure. In MH treated groups, MDA was significantly increased, with a maximum MDA level generated by PDHBH@MNC-mediated MH. Overall, data show that a limited oxidative stress was generated in the normal fibroblast cells by DHBH@MNC-mediated MH. Instead, colon cancer showed only minor oxidative stress, while in the A375 melanoma cells, the oxidative stress was enhanced, with the most important effect being generated by the DHBH@MNC-mediated MH. Therefore, in the fibroblasts and colon cancer cells, most of the ROS production seems to have been initiated by the lysosomal membrane permeabilization and to a lesser extent the MH-induced mitochondrial damage. The latter is responsible for the ROS generation in the melanoma cells, contributing significantly to the mechanism that induces apoptosis.

### 3.7. Western Blot

In the fibroblasts, p53 was inhibited following MH treatment, compared with controls, this effect was stronger for PDHBH@MNC (*p* = 0.0357, Kruskal–Wallis test) (Figure 12). BAX was increased by MNC internalization or MH treatment (not significant), the effect was stronger after MH treatment, the most powerful effect was induced by PDHBH@MNC-mediated MH (*p* = 0.0226, TTEST). BCL2 was increased by AMF exposure and PDHBH@MNC-mediated MH (not significant). Overall, the BAX/BCL2 ratio is increased by DHBH@MNC-mediated MH, leading to a pro-apoptotic effect, and it is decreased by PDHBH@MNC-mediated MH, favoring an anti-apoptotic effect, however, these alterations were not statistically significant. In the colon cancer CACO2 cells, there was an increase in the p53 level (not significant) following DHBH@MNC internalization and only small increases for the other groups. BAX was not significantly increased by AMF exposure and MH treatment, while BCL2 was increased by AMF exposure (*p* = 0.0125, TTEST) and decreased by MH treatment (*p* = 0.021 for PDHBH@MNC mediated MH, TTEST). This leads to a significantly altered BAX/BCL2 ratio (*p* = 0.0148, Kruskal–Wallis test), which favors apoptosis. In the A375 melanoma cells, p53 was strongly inhibited in all treated groups (*p* = 0.0199, Kruskal–Wallis test), MH treatment had a stronger effect. BAX was also inhibited by AMF exposure and MH treatment. The only significant increase was found for PDHBH@MNC. BCL2 was slightly decreased by AMF exposure but increased by internalized MNC w/wo MH treatment. Overall, the ratio BAX/BCL2 is decreased, although not significantly, favoring anti-apoptotic effects.

Kareem et al. [89] tested the effects of polyvinylpyrrolidone-loaded MnZnFe_2_O_4_ magnetic nanocomposites alone or as a combination therapy with NIR laser and alternating magnetic field (AMF) in the breast cancer cell lines AMJ-13 and MCF-7, the ovarian cancer cell line SKOV-3, as well as against the normal cell line HBL. Treated cancer cells with polyvinylpyrrolidone-loaded MnZnFe2O4 magnetic nanocomposites significantly increased ROS synthesis, with subsequent reduction of the mitochondrial membrane potential, resulting in apoptosis as a novel pathway that involves the mitochondrial damage mechanism via activated p53 [89]. This pathway seems to only partially explain the mechanism of the MH-induced cell death by the DHBH@MNC and PDHBH@MNC in our experiments. In our study, the levels of p53 and BAX/BCL2 ratio in melanoma were not increased compared with control, despite the increased levels of ROS—high caspase 3, decreased MMP and an association with the induction of efficient apoptosis—shown by FACS analysis. Most of the melanomas present a wild type (wt) oncogene suppressor p53, however, the p53 pathway is inactivated in 90% of cases [90], despite the low frequency, around 10% to 19%, of p53 disabling mutations [91]. However, in melanoma there is a specific inhibition of p53 transactivation of the pro-apoptotic promoters PIG3, BAX and PUMA. This leads to the loss of the pro-apoptotic role of p53 [92]. p53 activation leads to cell cycle arrest, apoptosis, and eventually growth inhibition of the tumor, therefore p53 activation is a key target in melanoma therapy, especially in combination with BRAF inhibitors, chemotherapy or radiotherapy [92].

In the colon cancer cell line, p53 was only significantly increased by PDHBH@MNC-mediated MH, associated with the downstream activation of the pro-apoptotic BAX and correlated with inhibition of the anti-apoptotic BCL2, leading to an efficient apoptosis induction. In the fibroblasts, there was an increase in the p53 level triggered by DHBH@MNC w/wo MH that correlated with BAX/BCL2 increase in the DHBH@MNC mediated MH, which was associated with increased ROS, suggestive of the mitochondrial pathway of apoptosis induction (increased caspase 9 and 3). However, in the tumor cells, the most important mechanism of cell death induction triggered by the DHBH@MNC and PDHBH@MNC-mediated MH seems to be membrane induced apoptosis, as seen by the significantly increased levels of caspase 8 and downstream activation of the caspase 3. The apoptosis induction is rather limited in the fibroblastic cell line, with DHBH@MNC-mediated MH having the stronger effect (around 10% of the total cell count) while maintaining a level below the toxicity limit. Meanwhile, in the tumor cell lines the PDHBH@MNC-mediated MH has a more pronounced cytotoxicity with the induction of cell death by the activation of both the extrinsic and also, to a lower extent, the intrinsic pathway. This is only a preliminary study, but the results are promising, since these synthesized nanoparticles, and especially PDHBH@MNC with better anti-tumor efficacy and increased biocompatibility on the normal fibroblastic cell line, can be further used for a combined therapy, such as MH and chemotherapy.

The major advantage of this study consists in the synthesis of the PDHBH@MNC, a nanoparticle that combines a biocompatible polymeric shell with different oligomeric units. This nanoparticle provides multiple reaction sites for the attachment of different therapeutic molecules (chemotherapeutic molecules) or tracers to create theranostic magnetic nanoplatforms [93]. Additionally, it can do this while retaining the enhanced magnetic properties capable of generating intracellular heat that leads to a selective apoptosis in two different cancer cell types. Obviously, both targeting capabilities and therapeutic effects can be improved by anchoring different functional molecules on the polymer scaffolds that encapsulate the magnetic core. These possibilities will be explored in future works.

Hyperthermia increases the efficiency of chemotherapy drugs because it increases the membrane permeability to the chemotherapy drugs. SPIONs added to OLN-93 rat glial and C6 rat glioblastoma cell lines confirm that the simultaneous exposure to AMF and temozolomide at various time intervals increased the number of apoptotic cells and the Bax/Bcl-2 mRNA ratio compared with temozolomide or AMF alone [94]. Combining antitumoral agents with MNPs, especially iron oxide nanoparticles (IONPs), seems a better antineoplastic strategy and an improved rationale to overcome multidrug resistance. Hence, a tandem apoptosis trigger (m-TAT) made of zinc-doped iron oxide MNPs, doxorubicin, and a death receptor 4 (DR4)-targeting monoclonal antibody (DR4 Ab) activates both extrinsic and intrinsic apoptosis signals and downregulation of the drug efflux pump of P-glycoprotein (P-gp), overcoming the MDR in both in vitro and in vivo studies [95]. Notable results were also found using PEG-modified lipid micelles loaded with SPIONs and sorafenib against hepatocellular carcinoma [96], PLGA–PEG coating iron oxide-docetaxel against breast cancer [97] and doxorubicin-loaded copolymer-coated SPIONs against colon cancer treatment [98].

An in vivo study with mice bearing metastatic breast cancer revealed that MH promotes the tumor-cell killing effect of radiotherapy through Bax-mediated cell death, improving cellular immunity in mice, which leads to augmented overall survival [99]. Another in vivo study in Ehrlich solid tumors revealed that MNPs coated with ascorbic acid showed a significant increase in the level of p53 expression [100]. Recent research has shown that the combination therapy using chitosan-coated MNPs (CS-MNPs) with 5-fluorouracil (5FU) and MH on a human colorectal cancer (HT29) heterotopic tumor model in mice increased the antitumor efficacy compared with the monotherapy [101].

## 4. Conclusions

We have demonstrated for the first time that 3,4-dihydroxybenzhydrazide (DHBH) and its polymer poly[3,4dihydroxybenzhydrazide] (PDHBH) can be efficiently used as stabilizers in the synthesis of magnetic nanoclusters with a multicore–shell architecture via solvothermal method. The use of PDHBH in the reaction mixture reduced the mean diameter by 40 nm with respect to DHBH-coated MNC. Moreover, the solvothermal method employed allows for consistent, reproducible results, which represents an important step into the transition towards clinical application. Both types of MNC are superparamagnetic at room temperature, exhibit high values of saturation magnetization and showed biocompatibility in vitro towards human and tumor cell lines. The AMF exposure led to a mild hyperthermia effect, which induced limited damage to the healthy cells and allows for selectivity of the therapeutic effect. In tumor cells, the magnetic hyperthermia treatment with both DHBH@MNC and PDHBH@MNC induced cell death by membrane pathway-triggered apoptosis and to a lower extent by mitochondrial pathway, the latter was mainly observed in melanoma cells. The PDHBH@MNC showed better antitumoral activity. PDHBH@MNC-mediated magnetic hyperthermia benefited from a biocompatible coating which offers further possibilities for attaching functional ligands, either for targeting or for chemotherapy, in view of further theranostic applications.

## Figures and Tables

**Figure 1 nanomaterials-13-00933-f001:**
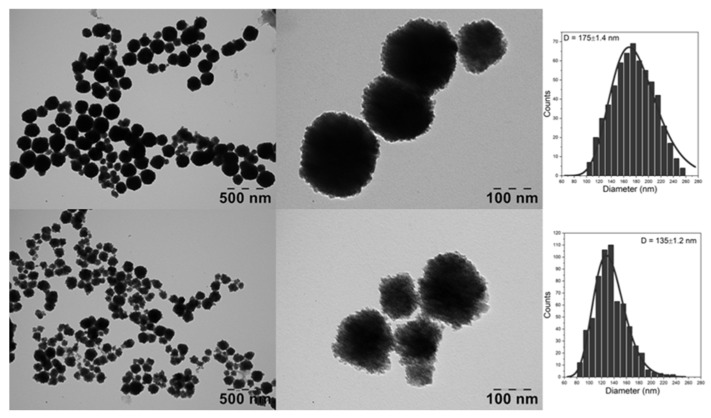
Large scale and high resolution TEM images together with the corresponding diameter distribution histograms fitted with a log-normal distribution (purple lines) of DHBH@MNC (upper panels) and PDHBH@MNC (lower panels).

**Figure 2 nanomaterials-13-00933-f002:**
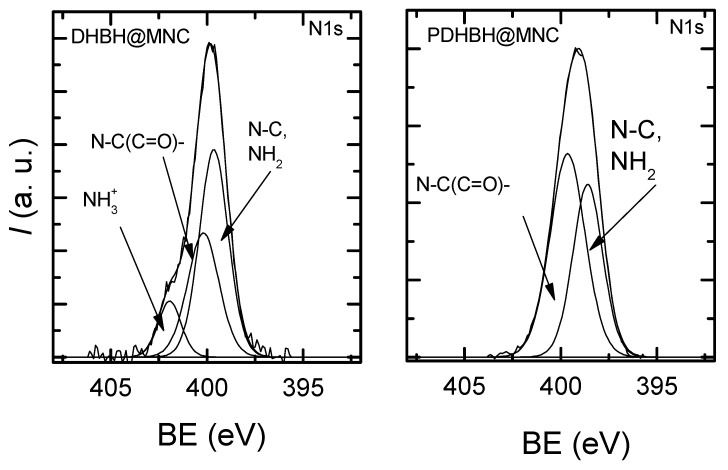
XPS N1s high resolution spectra obtained for DHBH@MNC and PDHBH@MNC.

**Figure 3 nanomaterials-13-00933-f003:**
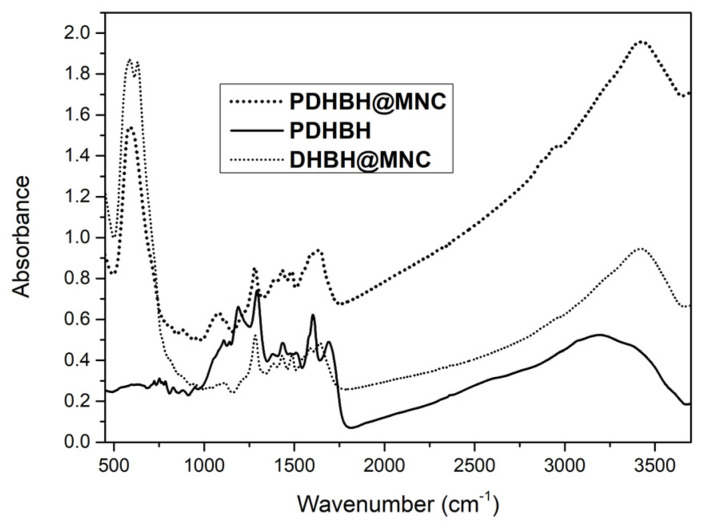
FT-IR spectra of PDHBH, PDHBH@MNC and DHBH@MNC.

**Figure 4 nanomaterials-13-00933-f004:**
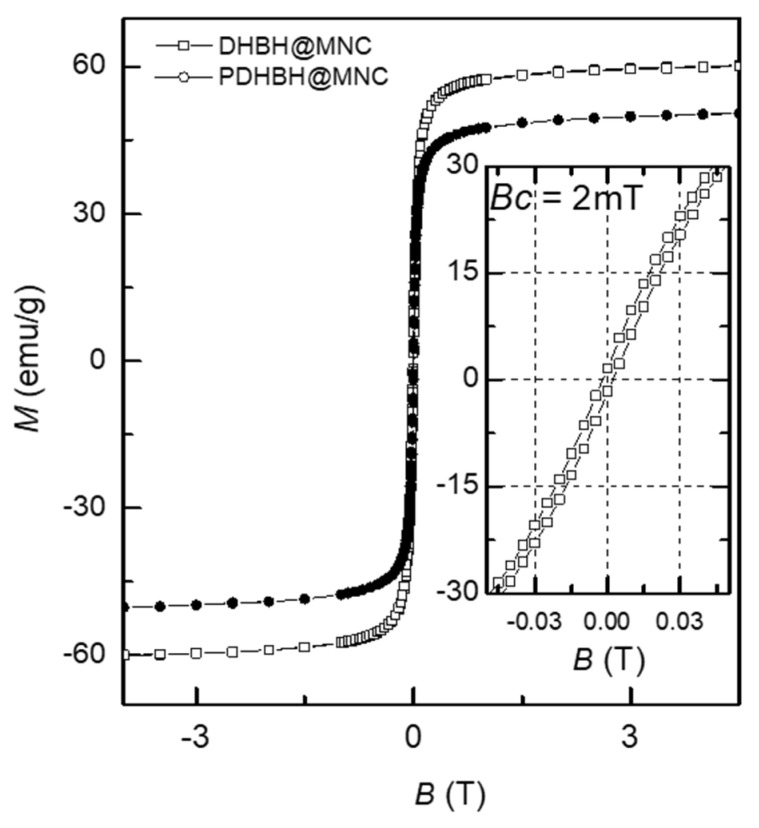
Magnetic hysteresis loops of both types of MNC. Inset is the low-field regime of the hysteresis loops.

**Figure 5 nanomaterials-13-00933-f005:**
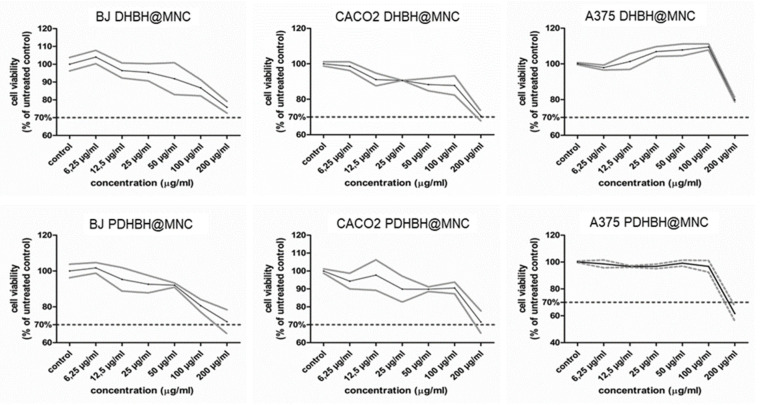
Intrinsic cytotoxicity of DHBH@MNC (upper panels) and PDHBH@MNC (lower panels) were tested against normal fibroblasts (BJ), colon cancer (CACO2) and human melanoma cells (A375) across a range of concentrations (0–200 µg/mL). Data are presented as % of untreated controls, (mean ± standard deviation, n = 3).

**Figure 6 nanomaterials-13-00933-f006:**
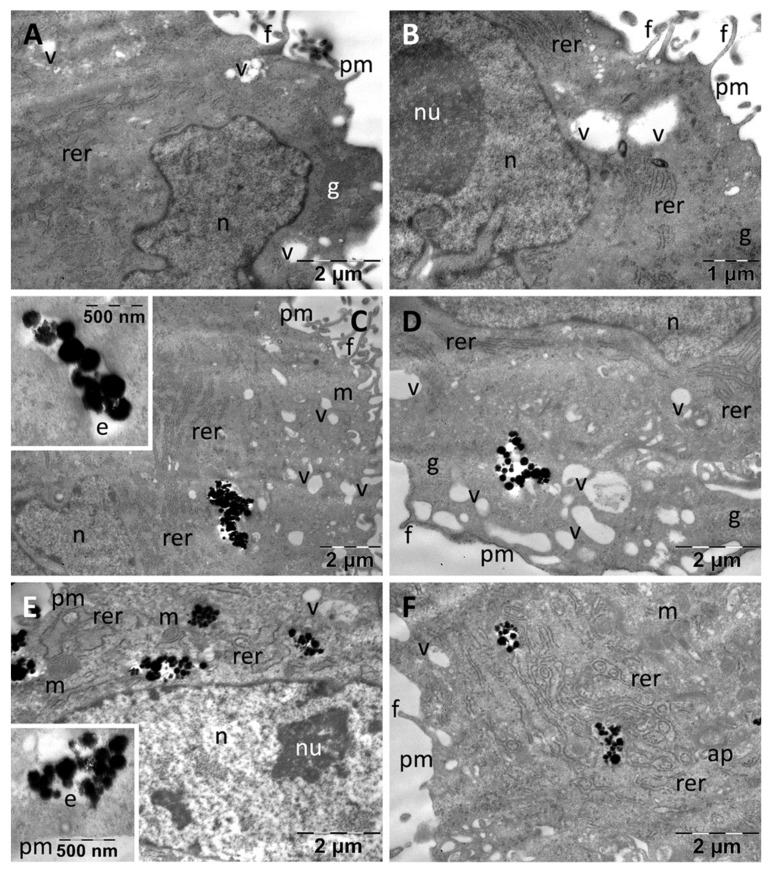
TEM images of BJ fibroblasts showing their normal ultrastructure in control group (**A**,**B**) and their internalization of the DHBH@MNC test group (**C**,**D**) and the PDHBH@MNC test group (**E**,**F**). ap: autophagosomes; e: endosomes; f: filopodia; g: glycogen; m: mitochondria; n: nucleus; nu: nucleolus; pm: plasma membrane; rer: rough endoplasmic reticulum; v: vacuoles.

**Figure 7 nanomaterials-13-00933-f007:**
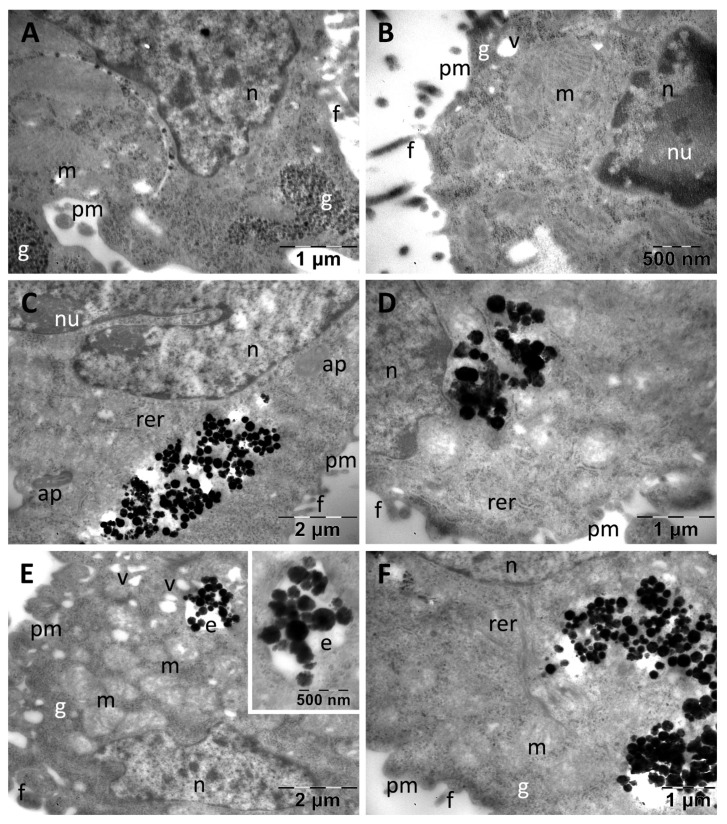
TEM images of CACO2 colon cancer cells showing their normal ultrastructure in control group (**A**,**B**) and internalization of the DHBH@MNC test group (**C**,**D**) and the PDHBH@MNC test group (**E**,**F**). ap: autophagosomes; e: endosomes; f: filopodia; g: glycogen; m: mitochondria; n: nucleus; nu: nucleolus; pm: plasma membrane; rer: rough endoplasmic reticulum; v: vacuoles.

**Figure 8 nanomaterials-13-00933-f008:**
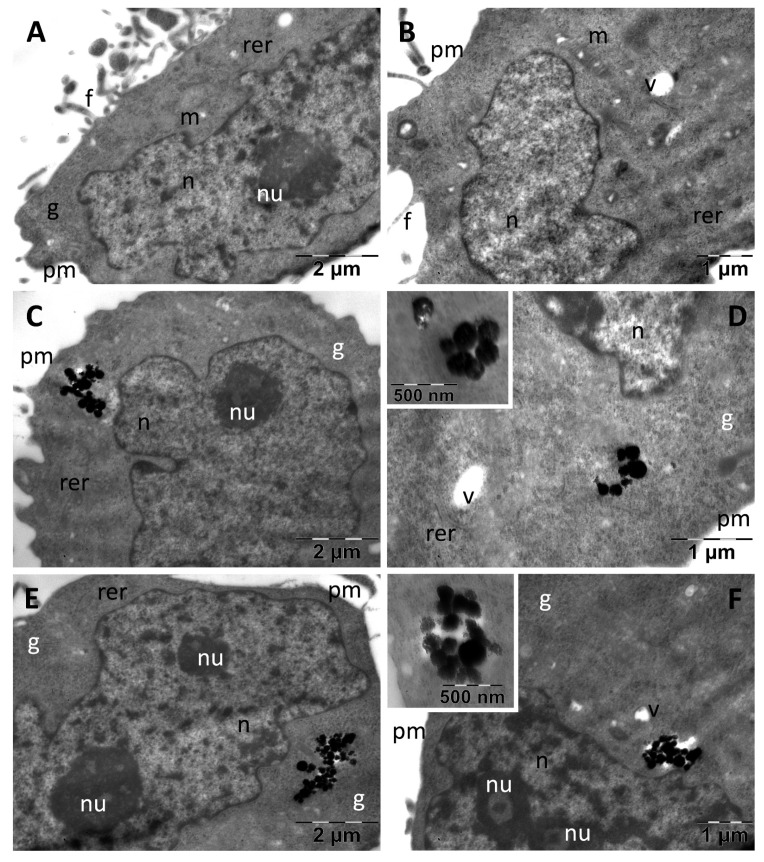
TEM images of A375 melanoma cells showing their normal ultrastructure in control group (**A**,**B**) and internalization of the DHBH@MNC test group (**C**,**D**) and the PDHBH@MNC test group (**E**,**F**). f: filopodia; g: glycogen; m: mitochondria; n: nucleus; nu: nucleolus; pm: plasma membrane; rer: rough endoplasmic reticulum; v: vacuoles.

**Figure 9 nanomaterials-13-00933-f009:**
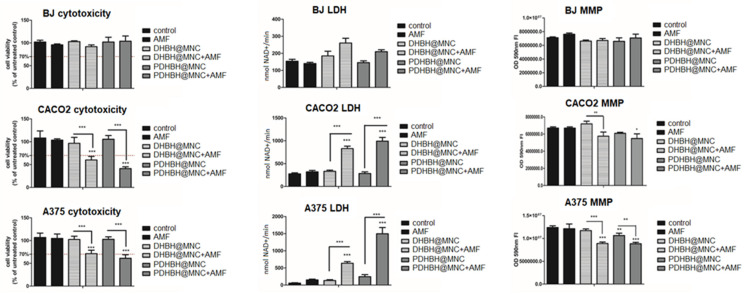
Cytotoxicity of cells (fibroblasts (BJ), upper panels; colon adenocarcinoma (CACO2), middle panel; and melanoma (A375), lower panel). Cells were exposed to AMF (frequency of 355Hz, amplitude of 40kA/m, time 40 min) and were incubated with either DHBH@MNC or PDHBH@MNC (50 µg/mL) with and without MH treatment. Cell cytotoxicity assay results are presented as % of untreated controls. LDH measurements are presented as nmolNAD+/min. Mitochondrial membrane potential alterations are measured by MITO ID assay; results are presented as fluorescence OD 590nm readings (right panels). Each bar represents mean ± standard deviation (n = 3); * = *p* < 0.05, ** = *p* < 0.001, *** = *p* < 0.0001, control versus treated group (cells exposed to AMF, incubated with MNC with and without MH exposure) for each cell line.

**Figure 10 nanomaterials-13-00933-f010:**
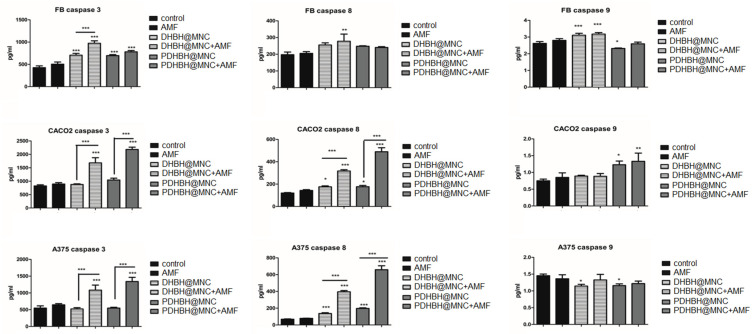
Caspase 3, 8, and 9 of cells (fibroblasts (BJ), upper panels; colon adenocarcinoma (CACO2), middle panel; and melanoma (A375), lower panel) measurements through ELISA. Results are expressed as pg/mL. Each bar represents mean ± standard deviation (n = 3); * = *p* < 0.05, ** = *p* < 0.001, *** = *p* < 0.0001, control versus treated group (cells exposed to AMF, incubated with MNC with and without MH exposure) for each cell line.

**Figure 11 nanomaterials-13-00933-f011:**
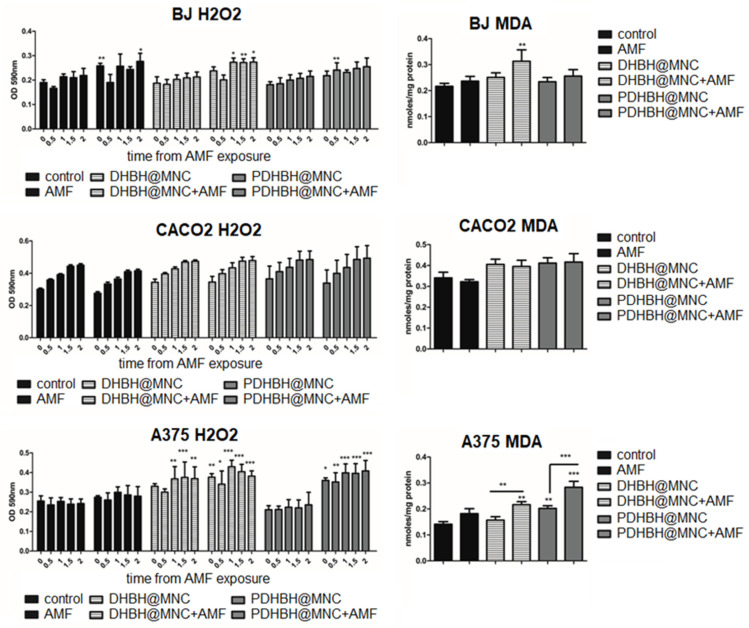
Oxidative stress of cells (fibroblasts (BJ), upper panels; colon adenocarcinoma (CACO2), middle panel; and melanoma (A375), lower panel) determined by Amplex Red measurement of the H2O2 level, immediately after MH treatment (readings were taken at 0, 0.5, 1, 1.5 and 2 h). Results are expressed as OD at 590nm (left panels) for each cell line; malondialdehyde (MDA) level at 24 h after MH treatment was measured through spectrophotometry, results are expressed as nmoles/mg protein (right panels). Each bar represents mean ± standard deviation (n = 3); * = *p* < 0.05, ** = *p* < 0.001, *** = *p* < 0.0001, control versus treated group (cells exposed to AMF, incubated with MNC with and without MH exposure) for each cell line.

**Figure 12 nanomaterials-13-00933-f012:**
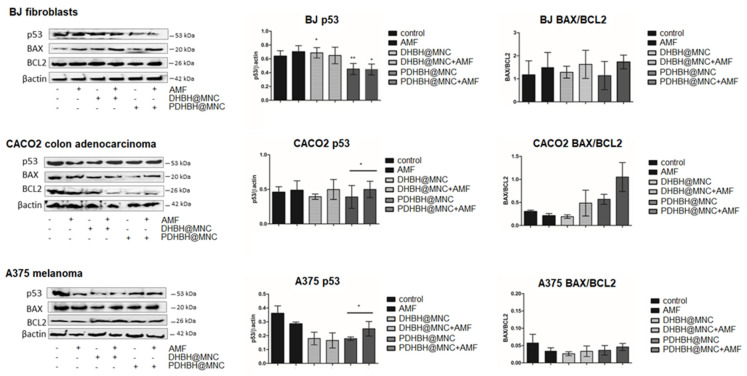
Western blot analysis of the protein expressions of the p53 apoptosis pathway of cells (fibroblasts (BJ), upper panels; colon adenocarcinoma (CACO2), middle panel; and melanoma (A375), lower panel). Image analysis of WB bands was undertaken by densitometry, results were normalized to β actin; BAX/BCL2 ratio is presented. Each bar represents mean ± standard deviation (n = 3); * = *p* < 0.05, ** = *p* < 0.001, control versus treated group (cells exposed to AMF, incubated with MNC with and without MH exposure) for each cell line.

## Data Availability

Not applicable.

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
