# Peer review of "Magnetic Nanoclusters Stabilized with Poly[3,4-Dihydroxybenzhydrazide] as Efficient Therapeutic Agents for Cancer Cells Destruction"

_nanomaterials, 2023, doi:10.3390/nano13050933_

Round 1

Reviewer 1 Report

The manuscript titled "Magnetic nanoclusters stabilized with poly[3,4-2 dihydroxybenzhydrazide] as efficient therapeutic agents for cancer cell destruction" presents an interesting discussion and a well-structured presentation. Overall, the manuscript shows novelty and has the potential for publication if the authors address my comments.

Firstly, the authors have provided a lengthy introduction, which is informative and engaging to read. However, it could benefit from a broader perspective of the field. Therefore, I suggest that the authors focus their discussion on the current topic and include more compelling biomedical applications of magnetic nanoparticles. For instance, the use of autofluorescence could be explored, as discussed in the Journal of Materials Chemistry C 10 (35), 12652-12679.

Secondly, the figures need to be consistent throughout the manuscript. The authors have used black and white figures in some instances, such as MH loops, while using color in others. I recommend that the authors use the same formatting and style for all the figures to maintain consistency.

Finally, while the manuscript presents several interesting results, some of them are merely observations that require further discussion. To improve the manuscript, I suggest that the authors expand on the results by adding more related physics, particularly in the discussion of each figure. An example of this can be found in the Journal of Alloys and Compounds 904, 163992, which discusses the FTIR results.

In conclusion, the manuscript is promising, and with revisions according to my comments, it could be considered for publication.

Author Response

Dear Reviewer 1,

Thank you for the careful reading of our Manuscript and for your positive evaluation. Your suggestions were indeed very useful for the increase of the paper quality and are incorporated in the new version of the manuscript as follows:

- we insert the suggested reference in the text: “Due to the bi-functionality of this type of nanostructures, they were successfully used for highly efficient labeling of human stem cell [8,9], as magnetic carrier for drug delivery [10-12], fluorescent intracellular probes [Kumar, 2022] as well as therapeutic agents in photodynamic therapy [13-15], and magnetic hyperthermia (MH) applications [16-21]”.

- the graphs were modified according to your suggestions so that in the new version of the manuscript we used the same formatting and style for all the figures to ensure consistency

- We added more explanations in the revised manuscript, as recommended by you, for FTIR, TEM and XPS results at Results and Discussions, sub-points 3.1 pages 6-8, with the proper references for similar magnetite materials.

Reviewer 2 Report

The authors synthesized magnetic core and polymer shell via in situ solvothermal process. They characterized the nanoclusters using TEM, XPS, VSM, FTIR. The authors also carried out magnetic hyperthermia (MH) treatment on tumor cells. They observed tumor cell apoptosis after MH treatments. The research is interesting and can be published to nanomaterials. I have a few comments.

(1)   In general, “nanoparticle” terminology is used. But “nanocluster” was used in the manuscript. Could you explain the reason?

(2)   The nanocluster mean diameter depended on polymers used: 175 nm for DHBH@MNC and 135 nm for PDHBH@MNC. Could you explain this result?

(3)   In Figure 4, M value of DHBH@MNC was higher than that of PDHBH@MNC. Could you explain this result?

(4)   The PDHBH@MNC showed better antitumoral activity than DHBH@MNC. Could you explain this result?

Overall, the manuscript is well-written and publishable to nanomaterials.

Author Response

Dear Reviewer 2,

thank you for the positive evaluation of our Manuscript. We will detail in our response below how we address each comment:

The authors synthesized magnetic core and polymer shell via in situ solvothermal process. They characterized the nanoclusters using TEM, XPS, VSM, FTIR. The authors also carried out magnetic hyperthermia (MH) treatment on tumor cells. They observed tumor cell apoptosis after MH treatments. The research is interesting and can be published to nanomaterials. I have a few comments.

  • In general, “nanoparticle” terminology is used. But “nanocluster” was used in the manuscript. Could you explain the reason?

Answer (1) :  The “nanoparticle” terminology is more general but “nanocluster” is the self-assembled structure of multiple nanoparticles by a specific chemical process, in our case solvothermal method. 

  • The nanocluster mean diameter depended on polymers used: 175 nm for DHBH@MNC and 135 nm for PDHBH@MNC. Could you explain this result?

Answer (2): We believe that in the case of using DHBH as stabilizer, even if we used 0.3g part of it remains unreacted or forms other species in the thermal process, which do not contribute in the clusterization process. Instead in the case of using PDHBH as stabilizer, being already a polymer, its structure is more compact and organized better the nanoparticles in the solvothermal process.

We detailed this answer in the text pg.6 lines 274-284

  • In Figure 4, M value of DHBH@MNC was higher than that of PDHBH@MNC. Could you explain this result?

Answer (3):  Generally, Ms is strongly dependent on the particle size [A.E. Berkowitz, W.J. Schuele, P.J. Flanders, J. Appl. Phys. 39, 1261 (1968)] since small nanoparticles have large surface area and thus the effect of spin disorder at the surface is high resulting in small Ms and high magnetic anisotropy. Therefore Ms value for DHBH@MNC (175nm) was higher than that of PDHBH@MNC (135nm). In addition, considering the core-shell structure, the surface layer of nanoclusters without magnetic ordering may also have an important contribution to the obtained results.

Additional explanations and bibliography were also included in the manuscript, pg.8, lines 312-326

  • The PDHBH@MNC showed better antitumoral activity than DHBH@MNC. Could you explain this result?

Answer (4): The answer is also related with points 2 and 3. It looks that already synthesized PDHBH used as stabilizer, has a different arrangement of the monomeric units in the polymeric matrix, part of it being oriented to the magnetite core and part to the external shell which is the interface with the external environment.  

Detailed text at pg.22 lines 693-701

Reviewer 3 Report

This manuscript deals with "Magnetic nanoclusters stabilized with poly[3,4-dihydroxybenzhydrazide] as efficient therapeutic agents for cancer cells destruction". This article claims that using of this type of magnetic nanoclusters could be a suitable for cancer therapy. Therefore, I suggest a minor correction and require a detailed clarification. Correction to be addressed by the authors as follows: The abstract is not well organized, where the sentences are incomplete and no continuity is there. It would be feasible, if include the significance of the current study in the abstract. A brief description of how the authors selected information from the literature in the databases, as well as what time period they searched for, is missing. Authors should justify and expand the information on the advantages of this study for biomedical applications. Authors should specify the main experimental conditions used on the evidences from the literature. Where they briefly describe the most important data reported in the literature in a homogeneous manner and sequence reinforcing the relevance of of this approach. Authors should discuss whether the use of this method represents a solid alternative to existing methods. Also please discuss about the role of mitochondria targeting. Please add below studies to your manuscript in discussion section using below manuscripts:

DOI: 10.1016/j.chemosphere.2022.134826

DOI: 10.3389/fchem.2021.674786

Conclusions should reaffirm the fundamental contribution of this paper.

Author Response

Dear Reviewer 3,

Thank you for the positive evaluation of our Manuscript. We will detail in our response below how we address each comment:

This manuscript deals with "Magnetic nanoclusters stabilized with poly[3,4-dihydroxybenzhydrazide] as efficient therapeutic agents for cancer cells destruction". This article claims that using of this type of magnetic nanoclusters could be a suitable for cancer therapy. Therefore, I suggest a minor correction and require a detailed clarification. Correction to be addressed by the authors as follows: The abstract is not well organized, where the sentences are incomplete and no continuity is there. It would be feasible, if include the significance of the current study in the abstract.

Answer: We reformulated the abstract structure:

Magnetic structures exhibiting large magnetic moments are sought after in theranostic approaches that combine magnetic hyperthermia treatment (MH) and diagnostic magnetic resonance imaging in oncology, since they offer an enhanced magnetic response to an external magnetic field. We report on the synthesis of core-shell magnetic structure two types of magnetite nanoclusters (MNC) based on a magnetite core and polymer shell, through an in-situ solvothermal process, using for the first time 3,4-dihydroxybenzhydrazide (DHBH) and poly[3,4-dihydroxybenzhydrazide] (PDHBH) as new stabilizers. Transmission Electron Microscopy (TEM) analysis showed formation of spherical MNC, X-ray photoelectonic spectroscopy (XPS) and Fourier Transformed-Infrared (FT-IR) analysis proved the existence of polymer shell. Magnetization measurement showed saturation magnetization values of 50 emu/g for PDHBH@MNC and 60 emu/g for DHBH@MNC with very low coercive field and remanence, indicating that the MNC are in a superparamagnetic state at room temperature, thus, being suitable for biomedical applications. MNCs were investigated in vitro, on human normal (dermal fibroblasts - BJ) and tumor (colon adenocarcinoma - CACO2, and melanoma - A375) cell lines, in view of toxicity, antitumor effectiveness and selectivity upon magnetic hyperthermia. MNCs exhibited good biocompatibility, and were internalized by all cell lines (TEM), with minimal ultrastructural changes. By means of flowcytometry apoptosis detection, fluorimetry, spectrophotometry for mitochondrial membrane potential, oxidative stress, ELISA-caspases, Western Blot-p53 pathway, we show that MH efficiently induced apoptosis mostly by membrane pathway and to a lower extent by mitochondrial pathway, the latter mainly observed in melanoma. Contrary, in fibroblasts the apoptosis rate was above the toxicity limit. Due to its coating, PDHBH@MNC showed selective antitumor efficacy and can be further used in theranostics since the PDHBH polymer provides multiple reaction sites for the attachment of therapeutic molecules.

A brief description of how the authors selected information from the literature in the databases, as well as what time period they searched for, is missing.

Answer:

The literature data was selected based on the similarity to our experimental setting: the synthesis and characterization methods for obtaining the iron oxide based magnetic nanoparticles, the AMF exposure conditions and the biological effects induced by MH in vitro, in vivo; clinical possible applications of such therapies. There was no time period limit for the literature assessment. Since this article is not a meta-analysis of the literature regarding the core-shell type magnetic nanoparticles, but an original research article, the literature data selection is based on discussing the background research in the field, as well as comparison between our findings and the existent data in the field.

Authors should justify and expand the information on the advantages of this study for biomedical applications.

Please add below studies to your manuscript in discussion section using below manuscripts:

DOI: 10.1016/j.chemosphere.2022.134826

DOI: 10.3389/fchem.2021.674786

We reformulated the text, pg.22 lines 693-701 and inserted one of the suggested references (Eftekhari A, Arjmand A, Asheghvatan A, Švajdlenková H, Šauša O, Abiyev H, Ahmadian E, Smutok O, Khalilov R, Kavetskyy T and Cucchiarini M (2021) The Potential Application of Magnetic Nanoparticles for Liver Fibrosis Theranostics. Front. Chem. 9:674786. doi: 10.3389/fchem.2021.674786), however we did not recommend citing the reference “Ahmadian E, Janas D, Eftekhari A, Zare N. Application of carbon nanotubes in sensing/monitoring of pancreas and liver cancer. Chemosphere. 2022 Sep;302:134826. doi: 10.1016/j.chemosphere.2022.134826. Epub 2022 May 4. PMID: 35525455”, since the topic had no connection with the aim of our manuscript related to magnetic nanoparticles or polydopamine analogues.

The major advantage of this study consists in the synthesis of the PDHBH@MNC a nanoparticle that combines a biocompatible polymeric shell with different oligomeric units that provide multiple reaction sites for the attachment of different others therapeutic molecules (chemotherapeutic molecules) or tracers to create theranostic magnetic nanoplatforms [93] while retaining the enhanced magnetic properties capable to generate intracellular heat that led to a selective apoptosis in two different cancer cell types. Obviously, both targeting capabilities and therapeutic effects can be improved by anchoring different functional molecules on the polymer scaffold that encapsulate the magnetic core. These possibilities will be explored in future works.

Authors should specify the main experimental conditions used on the evidences from the literature. Where they briefly describe the most important data reported in the literature in a homogeneous manner and sequence reinforcing the relevance of this approach.

In our experimental setting, the solvothermal method employed allowed for consistent, reproducible results, which represents an important step into the transition towards clinical application.  From a biomedical application point of view, the use of the PDHBH@MNC mediated hyperthermia benefited from a biocompatible coating which offers further possibilities for attaching functional ligands, either for targeting or for chemotherapy. The AMF exposure led to a mild hyperthermia effect, which induced a limited damage to the healthy cells, and allows for selectivity of the therapeutic effect. There are many in vitro and in vivo studies in the literature regarding the SPION mediated MH which show the induction of antitumor effects by the intracellular hyperthermia such as mithocondria or membrane apoptosis initiation, ROS generation, lysosomal membrane permeabilisation, effects that depend on the physical properties of the nanoparticles, such as the diameter, shape, magnetic properties or the concentration, intracellular localization, AMF exposure and the cell type reviewed in (69).

Also please discuss about the role of mitochondria targeting.

Pg.18 lines 524-547: Many mitochondria-targeted nanoparticles were synthesized for chemotherapy, photothermal therapy (PTT), photodynamic therapy (PDT), chemodynamic therapy (CDT), sonodynamic therapy (SDT), radiotherapy (RDT) and immunotherapy to enhance the anti-tumor efficacy (reviewed in Gao [78]). The aim was to trigger mitochondrial apoptosis and/or necrosis, by inducing damage to the mitochondrial DNA or oxidative stress. In PTT, the increase of the temperature in the mitochondrial area associated with the decrease of the heat shock proteins secretion can lead to mitochondrial cell death, while in PDT, SDT or RDT, the ROS generated by exposure to light, ultrasounds or ionizing radiation can lead to the same effect. There are many drawbacks of these procedures, such as treatment toxicity, low penetration in the solid tumors for PDT, or increased secretion of antiapoptotic factors that provide the cancer cells the means to escape therapy, that limit their efficacy. 

Gao Y, Tong H, Li J, Li J, Huang D, Shi J and Xia B (2021) Mitochondria-Targeted Nanomedicine for Enhanced Efficacy of Cancer Therapy. Front. Bioeng. Biotechnol. 9:720508. doi: 10.3389/fbioe.2021.720508

Pg.18 lines 536-547: In the apoptosis induction by the mitochondrial pathway, the mitochondrial permeability transition pore (PTP) protein complex plays a central role in the. PTP is located between the outer and the inner mitochondrial membrane and is mainly a voltage dependent anion channel at the outer membrane and an adenine nucleotide translocase at the inner membrane. PTP activation leads to increased permeability of the mitochondrial membranes, leading to the loss of the mitochondrial potential and subsequent large opening of the PTP to allow the cytochome c to be released into the cytoplasm and initiate caspaze activation. The process can be activated by BAX pro-apoptotic protein released by mild heat exposure (40°C) in bovine mammary epithelium [79]. Hyperthermia (43°C- exposure) was also shown to activate mitochondrial apoptosis in mice embryos [80], human osteosarcoma cells [81], however, in the latter, the cells also showed endoplasmic reticulum (ER) stress.

Juan Du, He-Shuang Di, Liang Guo, Zhong-Hao Li, Gen-Lin Wang,Hyperthermia causes bovine mammary epithelial cell death by a mitochondrial-induced pathway, Journal of Thermal Biology, Volume 33, Issue 1,2008, 37-47,https://doi.org/10.1016/j.jtherbio.2007.06.002

Kim WK, Mirkes PE. Alterations in mitochondrial morphology are associated with hyperthermia-induced apoptosis in early postimplantation mouse embryos. Birth Defects Res A Clin Mol Teratol. 2003 Nov;67(11):929-40. doi: 10.1002/bdra.10102.

Hou CH, Lin FL, Hou SM, Liu JF. Hyperthermia induces apoptosis through endoplasmic reticulum and reactive oxygen species in human osteosarcoma cells. Int J Mol Sci. 2014 Sep 29;15(10):17380-95. doi: 10.3390/ijms151017380.

Authors should discuss whether the use of this method represents a solid alternative to existing methods.

We are thankful for the reviewer’s suggestion and want to inform him that we already discussed in our manuscript about the solvothermal synthesis method (please check page 2, lines 75-77):

“Among different synthesis methods employed in the elaboration of MNC, the solvothermal / hydrothermal method is one of the most reproducible in terms of size or shape, by adjusting the reagent parameters [40–45]. “

Moreover, we also mentioned in the Conclusions section (page 23, lines 730-732) that the solvothermal method employed allows for consistent, reproducible results, which represent an important step into the transition towards clinical application. 

Conclusions should reaffirm the fundamental contribution of this paper.

Pg. 23, lines 726-743: We have demonstrated for the first time that 3,4-dihydroxybenzhydrazide (DHBH) and its polymer poly[3,4dihydroxybenzhydrazide] (PDHBH) can be efficiently used as stabilizers in the synthesis of magnetic nanoclusters with a multicore-shell architecture via solvothermal method. The use of PDHBH in the reaction mixture reduced the mean diameter by 40 nm with respect to DHBH coated MNC. Moreover, the solvothermal method employed allows for consistent, reproducible results, which represents an important step into the transition towards clinical application.  Both types of MNC are superparamagnetic at room temperature, exhibit high values of saturation magnetization and showed biocompatibility in vitro towards human and tumor cell lines. The AMF exposure led to a mild hyperthermia effect, which induced a limited damage to the healthy cells, and allows for selectivity of the therapeutic effect. In tumor cells, the magnetic hyperthermia treatment with both DHBH@MNC and PDHBH@MNC induced cell death by membrane pathway triggered apoptosis and to a lower extent by mitochondrial pathway, the latter was mainly observed in melanoma cells. The PDHBH@MNC showed better antitumoral activity. PDHBH@MNC mediated magnetic hyperthermia benefited from a biocompatible coating which offers further possibilities for attaching functional ligands, either for targeting or for chemotherapy, in view of further theranostic applications.

Round 2

Reviewer 1 Report

The manuscript can be accepted for publication.